# Deep Signature Statistics for Likelihood-free Time-series Models

Joel Dyer [1]    Patrick Cannon [2]    Sebastian M. Schmon [2]

## Abstract

Simulation-based inference (SBI) has emerged as a family of methods for performing inference on complex simulation models with intractable likelihood functions. A common bottleneck in SBI is the construction of low-dimensional summary statistics of the data. In this respect, time-series data, often being high-dimensional, multivariate, and complex in structure, present a particular challenge. To address this we introduce deep signature statistics, a principled and automated method for combining summary statistic selection for time-series data with neural SBI methods. Our approach leverages deep signature transforms, trained concurrently with a neural density estimator, to produce informative statistics for multivariate sequential data that encode important geometric properties of the underlying path. We obtain competitive results across benchmark models.

## 1. Introduction

In recent decades, scientific modelers have increasingly adopted *simulation-based models*: computer programs describing stochastic generative processes. Such models are widely employed in a variety of disciplines, e.g. economics (Baptista et al., 2016) and ecology (Wood, 2010). Their popularity lies in the greater flexibility afforded to the modeler over conventional equation-based modeling, enabling a higher degree of fidelity to the true data-generating process.

A drawback of this greater flexibility is that the likelihood functions of simulation models are typically intractable, being defined only implicitly (Diggle & Gratton, 1984). Consequently, traditional frequentist and Bayesian approaches relying on likelihood evaluations are infeasible. This limitation has motivated a plethora of methods for performing likelihood-free or simulation-based inference (SBI) (for a

recent overview, see Cranmer et al., 2020). Early examples of such methods include approximate Bayesian computation (ABC) (Pritchard et al., 1999; Beaumont et al., 2002) and synthetic likelihood (Wood, 2010), while more recent approaches exploit the flexibility of modern machine learning techniques, e.g. (sequential) neural likelihood estimation (Papamakarios et al., 2019; Lueckmann et al., 2019) and (sequential) neural ratio estimation (Hermans et al., 2020).

In traditional approaches, the selection of an appropriate, low-dimensional set of summary statistics is key to the quality of inference. A popular approach in ABC is to select a large set of candidate statistics from which lower dimensional summaries are obtained through 'best subset selection', 'projection' or 'regularization' (Blum et al., 2013). A major disadvantage of these approaches is that they require the user to know in advance a powerful set of summary statistics, which can be time-consuming and arbitrary and often requires domain knowledge and experimentation.

Some more recent approaches, such as sequential posterior and ratio estimation, are able to automate the learning of summary statistics by leveraging the expressiveness of neural networks. Yet, it is not guaranteed that the summary statistics implicitly generated through the use of such neural networks provide representations of sufficient quality for posterior inference. Fully connected networks and deep multilayer perceptrons (see e.g. Wong et al., 2018) still lack the inductive biases to easily extract meaningful representations from time-series, which poses the question of how automated techniques can fill the knowledge gap of domain expertise. One such possibility is partially exchangeable networks (PENs) (Wiqvist et al., 2019), in which a neural network architecture that exploits certain model symmetries is proposed in the context of ABC.

Here, we argue for the use of the so-called "signature method" (Morrill et al., 2020; Bonnier et al., 2019) for extracting features from multimodal, multivariate sequential data. The central object of study—the *path signature*—is, in a sense, a canonical feature transformation in that the signature of path-valued random variable captures all possible nonlinearities (Arribas, 2018). Applications of the signature method have produced promising results in a number of tasks, including character recognition (Xie et al., 2018), gesture recognition (Li et al., 2017), and early identifica-

---

[1]Mathematical Institute, University of Oxford [2]Improbable, London. Correspondence to: Joel Dyer <Joel.Dyer@maths.ox.ac.uk>.

Third workshop on *Invertible Neural Networks, Normalizing Flows, and Explicit Likelihood Models* (ICML 2021). Copyright 2021 by the author(s).

tion of Alzheimer's from clinical data (Moore et al., 2019). More recently, the signature has been proposed as a natural means for constructing distances between time series data in approximate Bayesian computation (Dyer et al., 2021).

We term our method deep signature statistics (DSS). The main idea is to embed a deep signature model (Bonnier et al., 2019)—in which the signature appears as a pooling operation in a neural network—into an existing neural method for posterior density estimation. Doing so combines the useful inductive biases provided by the signature transform with the power of neural networks to efficiently learn summary statistics and posterior estimates concurrently. Our results suggest that DSS offers a robust, automatic, and theoretically principled pipeline for posterior inference which performs competitively across benchmark models.

## 2. Background: Path signatures

The *signature* of a path $X = (X^1, X^2, \ldots, X^d): [0, T] \to \mathbb{R}^d$ is an infinite collection of statistics that characterizes the path up to a negligible equivalence class (Lyons, 2014). It is defined by the infinite collection of statistics

$$\mathrm{Sig}(X) = (1, S(X)_{0,T}^1, S(X)_{0,T}^2, \ldots, S(X)_{0,T}^d,$$
$$S(X)_{0,T}^{1,1}, S(X)_{0,T}^{1,2}, \ldots)$$

consisting of the $k$-fold iterated integral of $X$ with multi-index $i_1, \ldots, i_k$ defined as

$$S(X)_{0,T}^{i_1, \ldots, i_k} = \int \cdots \int_{0 \leq t_1 < \cdots < t_k \leq T} \mathrm{d}X_{t_1}^{i_1} \ldots \mathrm{d}X_{t_k}^{i_k}. \quad (1)$$

We provide a geometric interpretation of the depth 1 terms, $S(X)_{0,T}^i$, and depth 2 terms, $S(X)_{0,T}^{i,j}$, of the signature in Figure 1. When the underlying path $X$ is of bounded variation, the integral (1) can be understood as the Riemann-Stieltjes integral with respect to $X$. In particular, for differentiable paths, this leads to a more commonly understood Riemann integral by substituting $\mathrm{d}X_t = \frac{\mathrm{d}X_t}{\mathrm{d}t}\mathrm{d}t$.

$\mathrm{Sig}(X)$ can be understood as the equivalent of statistical moments for path-valued random variables, the terms of which constitute a set of "canonical features that can be intuitively described as an ordered version of sample moments" (Kiraly & Oberhauser, 2019). It is standard to truncate the infinitely long signature to depth $N \in \mathbb{N}$, which consists of all terms in the signature that have index sets $\{i_1, i_2, \ldots, i_k\}$ for $k \leq N$. We denote this with $\mathrm{Sig}_N(X)$. We further denote the set of all streams on a set $V$ by

$$\mathcal{S}(V) = \{\mathbf{x} = (x_1, \ldots, x_n): x_i \in V, n \in \mathbb{N}\}.$$

Here, to obtain a signature from a stream of data $\mathbf{x} \in \mathcal{S}(\mathbb{R}^d)$, the data points $x_i$ are first interpolated into a path before

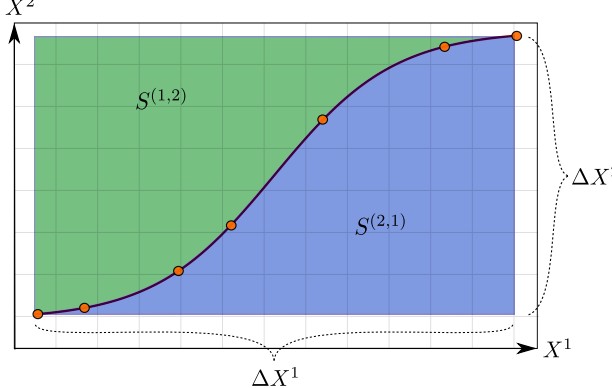

*Figure 1.* A geometric interpretation of signature terms. Orange circles indicate (possibly irregular) observations, and the black curve illustrates the underlying continuous path. Depth-1 terms are the increments $\Delta X^1$ and $\Delta X^2$, while the depth-2 terms $S^{(1,2)}$ and $S^{(2,1)}$ correspond to the green and blue areas, respectively.

the integrals of Equation 1 are computed. For example, one may use the interpolation via a continuous function $f = (f_1, \ldots, f_d): [0, T] \to \mathbb{R}^d$ with $f\left(\frac{i-1}{n-1} \cdot T\right) = x_i$ leaving us with the signature terms

$$S(X)_{0,T}^{i_1, \ldots, i_k} = \left(\int \cdots \int_{0 \leq t_1 < \cdots < t_k \leq T} \prod_{j=1}^{k} \frac{\mathrm{d}f_{i_j}}{\mathrm{d}t}(t_j)\mathrm{d}t_1 \cdots \mathrm{d}t_k\right).$$

## 3. Method: Deep Signature Statistics

A *deep signature transform* (Bonnier et al., 2019), shown in Figure 2, entails repeated application of *blocks* of three key elements: an augmentation of the stream with a stream-preserving feature map $\Phi^\varphi$ with learnable parameters $\varphi$; a lift operation $\ell$, which transforms the augmented stream into a stream of streams; and the depth $N$ signature transform applied to each substream, giving a stream of signatures. Blocks are by design able to concatenate, and after as many blocks as desired have been concatenated, the output is obtained by passing the output of the final block through an optional additional neural network. The ultimate effect is to capture higher order signature information using fewer terms (Chevyrev & Oberhauser, 2018; Kiŕaly & Oberhauser, 2019). We provide further details on deep signature transforms in Appendix A.

Combining path signatures—with their strong mathematical basis—with the expressivity of neural networks has been seen to produce competitive results in a number of learning tasks (Morrill et al., 2020). In this way, the use of a deep signature transform may yield approximately sufficient statistics, despite truncation. This makes it an ideal candidate for use in likelihood-free inference settings as a means for generating data-dependent summary statistics for both univariate and multivariate data of any length. We term

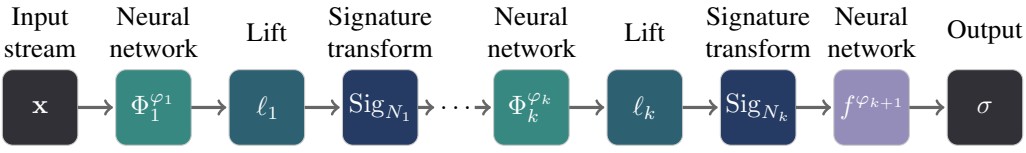

A single neural-lift-signature block

*Figure 2.* Deep signature transform with parameters $\varphi_1, \ldots, \varphi_{k+1}$.

deep signature transforms used in this way *deep signature statistics* (DSS).

We train DSS in tandem with a neural density estimator and using the same loss function. The SBI algorithm then targets the posterior $p(\boldsymbol{\theta} \mid \sigma^{\varphi}(\mathbf{x}))$ with a conditional density estimator $q_\phi(\boldsymbol{\theta} \mid \sigma^{\varphi}(\mathbf{x}))$, or a classifier $f(\boldsymbol{\theta}, \sigma^{\varphi}(\mathbf{x}))$, while the extended parameter set $(\phi, \varphi)$ is learned jointly using the loss function of the posterior density estimator. Throughout, we make use of neural ratio estimation (NRE) (Hermans et al., 2020) or its sequential version, written SNRE, using a ResNet classifier with batch size 50 and learning rate 0.005.

## 4. Experiments

### 4.1. Evaluation metrics

To assess the quality of the estimated posteriors, we compute the sliced Wasserstein distance (SWD) (Peyre & Cuturi, 2019) between samples from approximate ground truth posteriors and samples from the estimated posterior densities. In all cases, SWDs were computed using the Python Optimal Transport package (Flamary & Courty, 2017) and 1000 posterior samples from the posterior density estimated in each training round. To train the ratio estimator, we generate 1000 training examples during each round for 20 rounds.

### 4.2. Ornstein-Uhlenbeck process

The Ornstein–Uhlenbeck (OU) process (Uhlenbeck & Ornstein, 1930) is a proto-typical Gauss–Markov stochastic differential equation (SDE) model. We discretize the SDE such that the data $\mathbf{x} = (x_0, x_1, \ldots, x_T), x_i \in \mathbb{R}$ is generated according to

$$x_i = \theta_1 \exp(\theta_2)\Delta t + (1 - \theta_1 \Delta t)x_{i-1} + \frac{\epsilon_i}{2},$$

with $x_0 = 10$, $\Delta t = 0.2$, and $\epsilon_i \sim \mathcal{N}(0, \Delta t)$. The parameters $\boldsymbol{\theta} = (\theta_1, \theta_2)$ are to be inferred. We set uniform priors $\theta_1 \sim \mathcal{U}(0, 1)$ and $\theta_2 \sim \mathcal{U}(-2, 2)$, and generate ground truth observation $\mathbf{x}_o \sim p(\mathbf{x}|\boldsymbol{\theta}^*)$ at true parameter values $\boldsymbol{\theta}^* = (\theta_1^*, \theta_2^*) = (0.5, 1)$.

We plot the marginal posteriors obtained for the OU process using Metropolis-Hastings and DSS in Figure 3 We see from this that DSS + sequential neural ratio estimation (SNRE) is able to accurately recover the posterior density for this

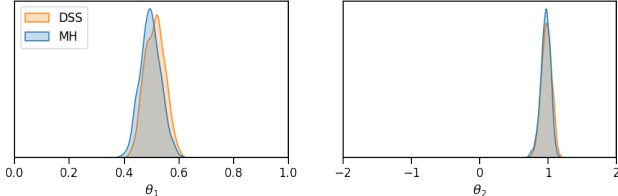

*Figure 3.* (Ornstein-Uhlenbeck) Example of the marginal posteriors obtained from DSS after round 10 (orange), and the approximate ground truth marginals from Metropolis-Hastings (blue).

model. A more quantitative evaluation of the quality of the estimated posteriors is shown in Figure 4, in which we compare the SWD between samples from the approximate ground truth posterior and estimated posteriors using each summary method at each training round. The hand-crafted summaries we used were the mean, standard deviation, and autocorrelations at lags 1 and 2 of the observed time series, giving four hand-crafted summary statistics.

Of the learned summaries, DSS tends to perform as well as or better than recurrent neural network (RNN) in 11 training rounds, while it outperforms PEN in almost all rounds. We also see for this simulator that the hand-crafted summary statistics outperform all learned summaries at almost every training round. This demonstrates the importance of well-chosen summary statistics and inductive biases, and the non-trivial nature of learning appropriate summary statistics for time series data: even with state-of-the-art neural network models such as RNN and PEN for summarizing time series data, it is difficult to meet, let alone surpass, the performance of sensible hand-crafted summaries.

### 4.3. Ricker model

The Ricker model (Ricker, 1954) is a simple ecological model of population dynamics with an intractable likelihood function. A population size $N_t$ evolves as

$$\log N_{t+1} = \log r + \log N_t - N_t + e_t,$$

where $r$ is a parameter determining the growth rate of the population and $e_t \sim \mathcal{N}(0, \sigma)$. The model assumes that the observations $\mathbf{x} = (x_0, x_1, \ldots, x_T), x_i \in \mathbb{R}$ are measurements of the population size, which in turn are Poisson random deviates $x_t \sim \text{Po}(\phi N_t)$, for scale parameter $\phi$. We assume the task of estimating the posterior

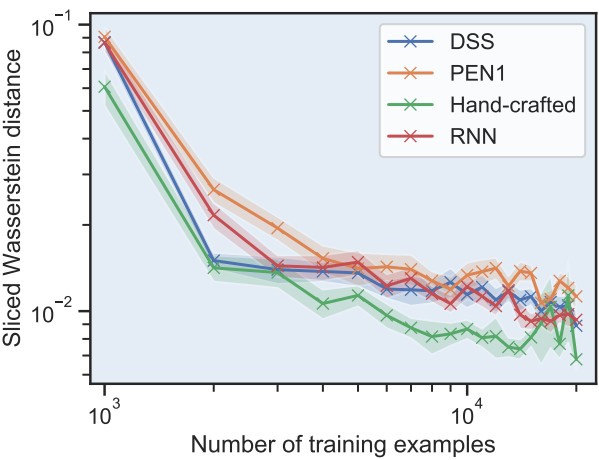

*Figure 4.* (Ornstein–Uhlenbeck) The sliced Wasserstein distances between approximate ground truth and estimated posterior densities for each summary method at each training round. Crosses and shaded regions indicate mean and standard error over 20 seeds.

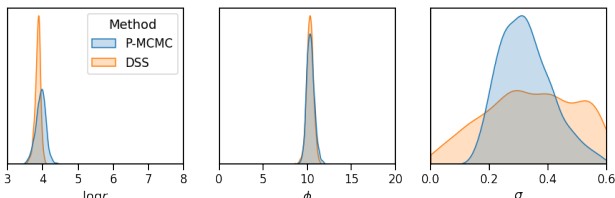

*Figure 5.* (Ricker model) An example of the marginal posteriors obtained from DSS after 10 rounds of 1000 (orange) and the approximate ground truth marginals from particle MCMC (blue).

density $p(\boldsymbol{\theta}|\mathbf{x})$ for $\boldsymbol{\theta} = (\log r, \phi, \sigma)$ given an observation $\mathbf{x}_o \sim p(\mathbf{x}|\boldsymbol{\theta}^*)$, where $\boldsymbol{\theta}^* = (4, 10, 0.3)$ is the true parameter set. We assume uniform priors for each parameter, with $\log r \sim \mathcal{U}(3,8)$, $\phi \sim \mathcal{U}(0,20)$, and $\sigma \sim \mathcal{U}(0,0.6)$.

In Figure 5, we plot samples from the approximate ground truth posterior $p(\boldsymbol{\theta}|\mathbf{x}_o)$—obtained using particle MCMC (Andrieu et al., 2010) following the guidelines of Schmon et al. (2020)—and the posteriors obtained using DSS for the Ricker model. From this, we see that DSS + SNRE has been reasonably successful in recovering the approximate ground truth density for $\sigma$, while it has accurately recovered the location and shape of the densities for $\log r$ and $\phi$.

In Figure 6, we show the SWD between the samples from the approximate ground truth posterior and estimated posteriors for each summary statistic method. The hand-crafted summary statistics used in this instance are those proposed in Wood (2010), and consist of: the autocovariances to lag 5; the mean; the number of zeros in the sequence; the coefficients of the regression $x_{t+1}^{0.3} = \beta_1 x_t^{0.3} + \beta_2 x_t^{0.6} + \epsilon_t$ for error term $\epsilon_t$; and the coefficients of the cubic regression of the ordered differences $x_t - x_{t-1}$ on their observed values.

From Figure 6, we see that DSS matches or exceeds PEN's

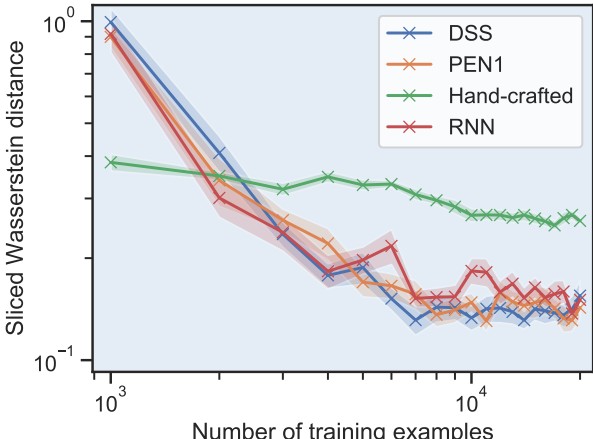

*Figure 6.* (Ricker model) The sliced Wasserstein distances between the true and estimated posterior densities for each summary statistic method at each training round. Crosses and shaded regions indicate mean and standard error over 20 different seeds.

(resp. RNN's) performance in 16 (resp. 18) out of 20 rounds. In particular, DSS appears to achieve greater asymptotic accuracy of the recovered posteriors at a high number of training examples. This example also highlights the possibility that learned summary statistics can outperform expert hand-crafted summaries, in particular, when model complexity doesn't allow for straightforward selection.

## 5. Discussion

In this paper, we address the problem of learning summary statistics for implicit time-series models with intractable likelihood functions. We propose the use of path signatures as a means for automatically generating approximately sufficient statistics for general multivariate time-series data. We demonstrate how the truncated signature can be combined with neural networks via deep signature transforms to generate informative summaries, and observe competitive performance in comparisons against existing state-of-the-art methods. Our method is general and non-specific to the models we consider. In particular, the size of the learned statistic can be taken to be model-dependent, while in our experiments the summary statistics learned with DSS are always of size 3. Furthermore, while we use the signature truncated to degree 3 in each neural-lift-signature block; this can be replaced with the log-signature truncated to a higher degree if more signature terms are required, while keeping the dimensionality low.

ACKNOWLEDGEMENTS

The authors thank James Morrill and Horatio Boedihardjo for helpful discussions. JD is supported by the EPSRC Centre For Doctoral Training in Industrially Focused Mathematical Modelling (EP/L015803/1) in collaboration with Improbable.

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

# Supplementary Material

# A. Deep Signature Transforms

We now describe the components of deep signature models in detail.

## STREAM-PRESERVING FEATURE MAP

The learnable, stream-preserving neural network $\Phi^\varphi : \mathbb{R}^{d \times m} \to \mathbb{R}^e$ for some $m \in \mathbb{N}$ operates on the original stream $\mathbf{x}$ as

$$\Phi(\mathbf{x}) = (\Phi_1, \ldots, \Phi_{n-m+1}),$$

where $\Phi_k = \Phi^\varphi(x_k, \ldots, x_{k+m}; \Phi_{k-1})$ and $\Phi_0 = 0$. This general structure can take the form of a one-dimensional convolutional layer, a feedforward network, or recurrent network.

## LIFT OPERATION

The learnable feature map obtained from the stream-preserving neural network augments the existing stream with additional channels. Its operation is described as "stream-preserving" since it does not destroy the stream-like nature of the data. The signature transform, on the other hand, operates on streams to produce an infinite set of features with no inherent stream-like properties. Direct application of the signature transform will thus prohibit its further application.

In general, however, we may wish to apply the signature transform repeatedly. This motivates the inclusion of a lift operation between the learnable, stream-preserving network and the signature transformation. A lift operation $\ell : \mathcal{S}(\mathbb{R}^d) \to \mathcal{S}(\mathcal{S}(\mathbb{R}^e))$ for some $e \in \mathbb{N}$ maps a stream into the space of streams of streams. Applying the signature transform element-wise to the lifted stream therefore yields a stream of signatures,

$$\operatorname{Sig}_N(\ell(\mathbf{x})) := (\operatorname{Sig}_N(\ell_1(\mathbf{x})), \ldots, \operatorname{Sig}_N(\ell_v(\mathbf{x})))$$
$$\in \mathcal{S}(\mathbb{R}^{(e^{N+1}-1)/(e-1)}),$$

which is amenable to further signature-based analysis (because the output is a stream). Examples of a lift operation include expanding windows $\ell(\mathbf{x}) = (\mathbf{x}_2, \mathbf{x}_3, \ldots, \mathbf{x}_n)$ where $\mathbf{x}_i = (x_1, \ldots, x_i)$, or sliding windows with window length $p$, in which case $\mathbf{x}_i = (x_i, \ldots, x_{i+p})$.

## NEURAL-LIFT-SIGNATURE BLOCK

A stream-preserving neural network can be combined with a lift-signature operation to create a neural-lift-signature block

$$B_N^\varphi(\mathbf{x}) = (\operatorname{Sig}_N \circ \ell \circ \Phi^\varphi)(\mathbf{x}).$$

This composite operation may or may not be stream-preserving. In particular, a neural-lift-signature block is not stream-preserving if we take $\ell(\mathbf{x}) := \mathbf{x}$ for that block.

## DEEP SIGNATURE TRANSFORMS

Let $\mathcal{X}$ be some set and $f^\varphi : \mathcal{S}(\mathbb{R}^c) \to \mathcal{X}$ be a neural network with trainable parameters $\varphi$. A deep signature transform $\sigma(\mathbf{x})$, illustrated in Figure 2, is a mapping from $\mathcal{S}(\mathbb{R}^d)$ to $\mathcal{X}$ defined as any sequence of $k$ neural-lift-signature blocks followed by an optional final neural network $f^{\varphi_{k+1}}$, i.e.

$$\sigma^\varphi(\mathbf{x}) = \left(f^{\varphi_{k+1}} \circ B_{N_k}^{\varphi_k} \circ \cdots \circ B_{N_2}^{\varphi_2} \circ B_{N_1}^{\varphi_1}\right)(\mathbf{x}) \quad (2)$$

where $\varphi = (\varphi_1, \ldots, \varphi_{k+1})$. Note that the lift operation can be different in each of the $k$ neural-lift-signature blocks $B_{N_k}^{\varphi_k}$.

# B. Implementation details

## B.1. Software

For evaluating signatures and deep signature transforms, we used `iisignature` (Reizenstein & Graham, 2020) and `https://github.com/patrick-kidger/Deep-Signature-Transforms` (Bonnier et al., 2019). SBI algorithms were implemented using `sbi` (Tejero-Cantero et al., 2020). The python implementation of PEN is found at `https://github.com/LoryPack/SM-ExpFam-LFI` (Pacchiardi & Dutta, 2020).

## B.2. Neural network specifications

### B.2.1. DEEP SIGNATURE STATISTICS

The deep signature model we use involved three neural-lift-signature blocks followed by a final recurrent network. The neural component of the first block consisted of a feedforward network with kernel size 3 and 2 hidden layers of size 16 swept across the input stream. The output size of this network was 3, so that initial layer augmented the input stream with an additional 3 channels. The neural components of the remaining two blocks were recurrent networks with 2 hidden layers of size 16. For each block, we use expanding windows with initial size 2 that grew by 1 time step in each iteration, followed by the signature transform truncated at degree 3. For all simulators, we apply basepoint and time augmentations to the input stream before passing it through the deep signature model, and take an output of size 3. This yields a model with 9,735 trainable parameters.

### B.2.2. PARTIALLY EXCHANGEABLE NETWORKS

Let $\mathbf{x} = (x_1, \ldots, x_n), x_i \in \mathcal{X}$ be sequential data generated by a stochastic process of Markov order $r$, and $A$ be a metric

space. Partially exchangeable network models $F : \mathcal{X}^n \to A$ consist of two networks $\phi : \mathcal{X}^{r+1} \to \mathbb{R}$ and $\rho : \mathcal{X}^r \times \mathbb{R} \to A$ combined as

$$F(\mathbf{x}) = \rho \left( x_{1:r} \sum_{i=1}^{n-r} \phi \left( x_{i:(i+r)} \right) \right).$$

For our experiments, we follow Wiqvist et al. (2019) and take the $\phi$ network to be a fully connected network with three layers of sizes 11, 100, and 50 and output size 10, and the $\rho$ network to be a fully connected network with four layers of sizes $(10 + r)$, 50, 50, and 20. ReLU activations were used for all hidden layers. For PEN1, this yields a model with 10,093 trainable parameters.

### B.2.3. RECURRENT NEURAL NETWORK

The recurrent network model consists of two recurrent neural networks. The first network has layers of size 64, 64, and 32, with an output of size 6, while the second layer has layers of size 32, 32, and 32 with output size 7. Windows of size 4 were swept across the input for both networks, with strides of 4 and 2 in the first and second, respectively. Altogether, this yields a model with 10,157 trainable parameters.