# OpenReview forum: "Deep Signature Statistics for Likelihood-free Time-series Models"
_ICML.cc/2021/Workshop/INNF — INNF+ 2021 poster_

### Official Review · Reviewer_sLZG · 2021-06-09

**Rating:** Borderline Reject
**Confidence:** 2

**Summary:**

In this paper, the authors propose deep signature statistics (DSS), which combines deep signature transform and neural ratio estimation to do posterior estimation on time series data. In the experiments, the proposed method is applied on two models, i.e.,  Ornstein-Uhlenbeck process, and  Ricker model.

**Justification For Rating:**

In my opinion, the paper has two flaws.

1. I am a little concerned about the novelty of DSS. It is only a direct application of deep signature transform.

2. The authors state in previous sections that the model can model high-dimensional and multivariate time series data. But in the experiments, the authors only test the model on simple statistical models. Specifically, the data is one-dimensional, i.e., $x_i \in R$. I am curious about if the model can be applied to more complex data.

---

### Official Review · Reviewer_F5DY · 2021-06-11

**Rating:** Accept
**Confidence:** 3

**Summary:**

This paper incorporates a learnable feature extractor into signature statistics for time series (path) data. Experimental evidence on several toy models (two SDEs) demonstrates that the learned signature statistics outperform hand-crafted signature statistics. Moreover, relative to other likelihood-free methods, the proposed method is better able to match the ground-truth posterior on these toy models, as measured in the (sliced) Wasserstein distance.

**Justification For Rating:**

Good workshop paper, the authors did a good job explaining their method in the limited space available. For a workshop paper, experimental evidence is convincing.

I am curious how the method performs on real-world high dimensional time series data (where the underlying process is unknown).

Minor comments:
-- Fig 6 y-axis is missing enough information to infer scale. Please include more ticks on the axis.
-- It would be good to include the information of Figures 5 & 6 in a table form. Ie, so that future work can use the sliced Wasserstein distances  in future comparisons.

---

### Decision · Program_Chairs · 2021-06-14

Accept (poster)